Resource

# Probing the limitations of multimodal language models for chemistry and materials research

Nawaf Alampara [1], Mara Schilling-Wilhelmi [1], Martiño Ríos-García [1], Indrajeet Mandal [2], Pranav Khetarpal [3], Hargun Singh Grover[4], N. M. Anoop Krishnan [2,3,4] ✉ & Kevin Maik Jablonka [1,5,6,7] ✉

Recent advancements in artificial intelligence have sparked interest in scientific assistants that could support researchers across the full spectrum of scientific workflows, from literature review to experimental design and data analysis. A key capability for such systems is the ability to process and reason about scientific information in both visual and textual forms—from interpreting spectroscopic data to understanding laboratory set-ups. Here we introduce MaCBench, a comprehensive benchmark for evaluating how vision language models handle real-world chemistry and materials science tasks across three core aspects: data extraction, experimental execution and results interpretation. Through a systematic evaluation of leading models, we find that although these systems show promising capabilities in basic perception tasks—achieving near-perfect performance in equipment identification and standardized data extraction—they exhibit fundamental limitations in spatial reasoning, cross-modal information synthesis and multi-step logical inference. Our insights have implications beyond chemistry and materials science, suggesting that developing reliable multimodal AI scientific assistants may require advances in curating suitable training data and approaches to training those models.

The practice of science has always required assimilating and integrating diverse forms of information, from visual observations in the laboratory and measurements to theoretical frameworks and previous literature. Although automation has traditionally excelled at repetitive tasks such as high-throughput experimentation[1–4], capturing the fundamental characteristic of scientific work—that is, the ability to interpret and connect multiple modes of information flexibly—has remained a central challenge for scientific discovery.

Recent advances in artificial intelligence, particularly in large language models (LLMs), have sparked renewed interest in developing more flexible computational systems for scientific workflows. These models can orchestrate specialized tools and combine general reasoning capabilities with domain-specific functions, suggesting a path towards more adaptable scientific automation[5–11]. However, a fundamental challenge persists: bridging the gap between human scientists' natural ability to seamlessly integrate visual, numerical

¹Laboratory of Organic and Macromolecular Chemistry (IOMC), Friedrich Schiller University Jena, Jena, Germany. ²School of Interdisciplinary Research, Indian Institute of Technology Delhi, Hauz Khas, New Delhi, India. ³Department of Civil Engineering, Indian Institute of Technology Delhi, Hauz Khas, New Delhi, India. ⁴Yardi School of Artificial Intelligence, Indian Institute of Technology Delhi, Hauz Khas, New Delhi, India. ⁵Center for Energy and Environmental Chemistry Jena (CEEC Jena), Friedrich Schiller University Jena, Jena, Germany. ⁶Helmholtz Institute for Polymers in Energy Applications Jena (HIPOLE Jena), Jena, Germany. ⁷Jena Center for Soft Matter (JCSM), Friedrich Schiller University Jena, Jena, Germany. ✉e-mail: krishnan@iitd.ac.in; mail@kjablonka.com

and textual information and the current limitations of computational systems in processing these different data types. This gap becomes particularly apparent in tasks that require combining visual interpretation with scientific reasoning, such as analyzing spectroscopic data[12], interpreting experimental set-ups[13] or evaluating safety conditions in laboratories[14,15].

Recent work has shown promising capabilities of LLMs in scientific tasks, from literature mining[16–23] and property prediction[10,24–30] to experiment planning[31–34]. Similarly, vision large language models (VLLMs) have demonstrated increasing capabilities in general visual reasoning tasks[35–39]. Although recent benchmarks have evaluated either the scientific reasoning capabilities of language models[40,41] or general multimodal abilities[35,36,42,43], a systematic evaluation of how these models handle the interplay of different modalities across the entire scientific process has been missing. This raises a crucial question: what are the limits of these models as co-pilots accelerating materials and chemistry research involving multimodal information extraction, simulations or experiments, and data analysis? Although we have some understanding for text-only LLMs, we still have no understanding for VLLMs that can process images alongside text.

To address this gap, we present the materials and chemistry benchmark (MaCBench), a comprehensive benchmark that evaluates multimodal capabilities across three fundamental pillars of the scientific process: information extraction from the literature, experimental execution and data interpretation. By focusing on these pillars, we can assess models' abilities across the full spectrum of scientific tasks, from understanding published results to executing and interpreting new experiments. Our benchmark is distinctively designed to not only measure performance but also to uncover the underlying failure modes of current models systematically. Through carefully constructed ablation studies, we investigate how performance varies across different modalities, levels of domain expertise required, reasoning complexity, and the distance to the training data corpus. This systematic approach allows us to test the hypothesis that current models might rely on superficial pattern matching rather than deeper scientific understanding. Our results reveal that although models can handle certain modalities individually, they often fail when tasks require flexible integration of information types—a core capability required for scientific work. For instance, models might correctly perceive information but struggle to connect these observations in scientifically meaningful ways.

These insights have important implications for developing AI-powered scientific assistants and self-driving laboratories. Our results highlight the specific capabilities needing improvement for these systems to become reliable partners in scientific discovery. They also suggest that fundamental advances in multimodal integration and scientific reasoning may be needed before these systems can truly assist in the creative aspects of scientific work.

## Results

### The MaCBench framework

Our benchmark design is guided by the observation that scientific work requires not only access to multiple modalities of information but also the ability to flexibly integrate them. To probe these capabilities of VLLMs meaningfully—rather than creating artificial question–answer-based challenges—we focus on tasks that mirror real scientific workflows, from interpreting scientific literature to evaluating laboratory conditions and analyzing experimental data (see Fig. 1). This approach allows us to evaluate the models' ability to process different types of information and their capacity to use this information to support scientific discovery. To assess performance in a broad range of settings, we rely on both images we mined from patents but also some we generated from scratch.

The benchmark is structured around three key aspects that form the basis of many scientific workflows: information extraction, in silico

or laboratory experiments, and data interpretation. Within each pillar, we include tasks spanning various scientific activities (see Fig. 2). The information extraction pillar analyzes the performance in parsing scientific literature, including extracting data from tables and plots, and interpreting chemical structures. The experiment execution pillar evaluates the models' ability to understand laboratory safety, identify equipment, assess safety conditions and understand crystal structures (as potential simulation artifacts). The data interpretation pillar tests models' capability to analyze various types of scientific data, from spectral analysis to electronic structure interpretation.

Here, a task refers to a single prompt template containing multiple questions. A task can either be a multiple-choice question (MCQ) or a numeric-answer question. The current corpus has 779 MCQs and 374 numeric-answer questions. A topic is a collection of tasks related to the same topic (one topic can have different types of tasks related to that topic; for example, X-ray diffraction (XRD) can have multiple tasks related to identifying peak positions, and then another set of tasks related to ordering peak positions in ascending/descending order). The three overarching focus areas are data extraction, data interpretation and experiments, each encompassing multiple topics.

### Performance landscape

There is considerable variation in model performance across different task types and modalities (Fig. 3; see Supplementary Table 1 for detailed descriptions of all tasks); however, when averaged over different tasks, Claude 3.5 Sonnet is the leading model on all three task families. Notably, the models do not fail at one specific part of the scientific process but struggle in all of them, suggesting that broader automation is not hindered by one bottleneck but requires advances on multiple fronts. Interestingly, even for a foundational pillar of the scientific process—that is, data extraction—some models do not perform much better than random guessing (for instance, Llama 3.2 90B Vision in Fig. 3). Current systems tend to perform best on MCQ-based perception tasks (for example, laboratory equipment and hand-drawn molecules in Fig. 3).

**Data extraction.** Our analysis shows that the first step of the scientific workflow, that is, data extraction, already poses considerable challenges for the models we tested. This is particularly the case when extracting science-specific data, for instance, on organic reactions and molecules. Although the best models perform well at extracting information on reaction diagrams, they fail to correctly describe the relationship between isomers (see Supplementary Fig. 4). As discussed below, this is probably caused by models struggling with spatial reasoning. Furthermore, even the extraction of compositions from tables still shows room for improvement for the VLLMs we tested (average accuracy of 0.53), performing indistinguishably from random guessing for Llama 3.2 90B Vision.

**In silico and laboratory experiments.** A similar variance in performance is observed for tasks related to the execution of laboratory or in silico experiments. Although models show good performance at recognizing laboratory equipment (average accuracy of 0.77), reasoning about laboratory scenarios, for example, comparing the safety hazards of two similar laboratory set-ups, exhibits low performance (average accuracy of 0.46).

The disparity between equipment identification and safety assessment performance suggests that although models can learn to recognize standard laboratory equipment, they still struggle with the more complex reasoning required for safe laboratory operations, questioning their ability to assist in real-world experiment planning and execution. This finding also implicates that current models cannot bridge gaps in tacit knowledge frequently discussed in biosafety scenarios[44,45].

Furthermore, the interpretation of crystal structure renderings—a crucial step for in silico experiments—demonstrates performance that

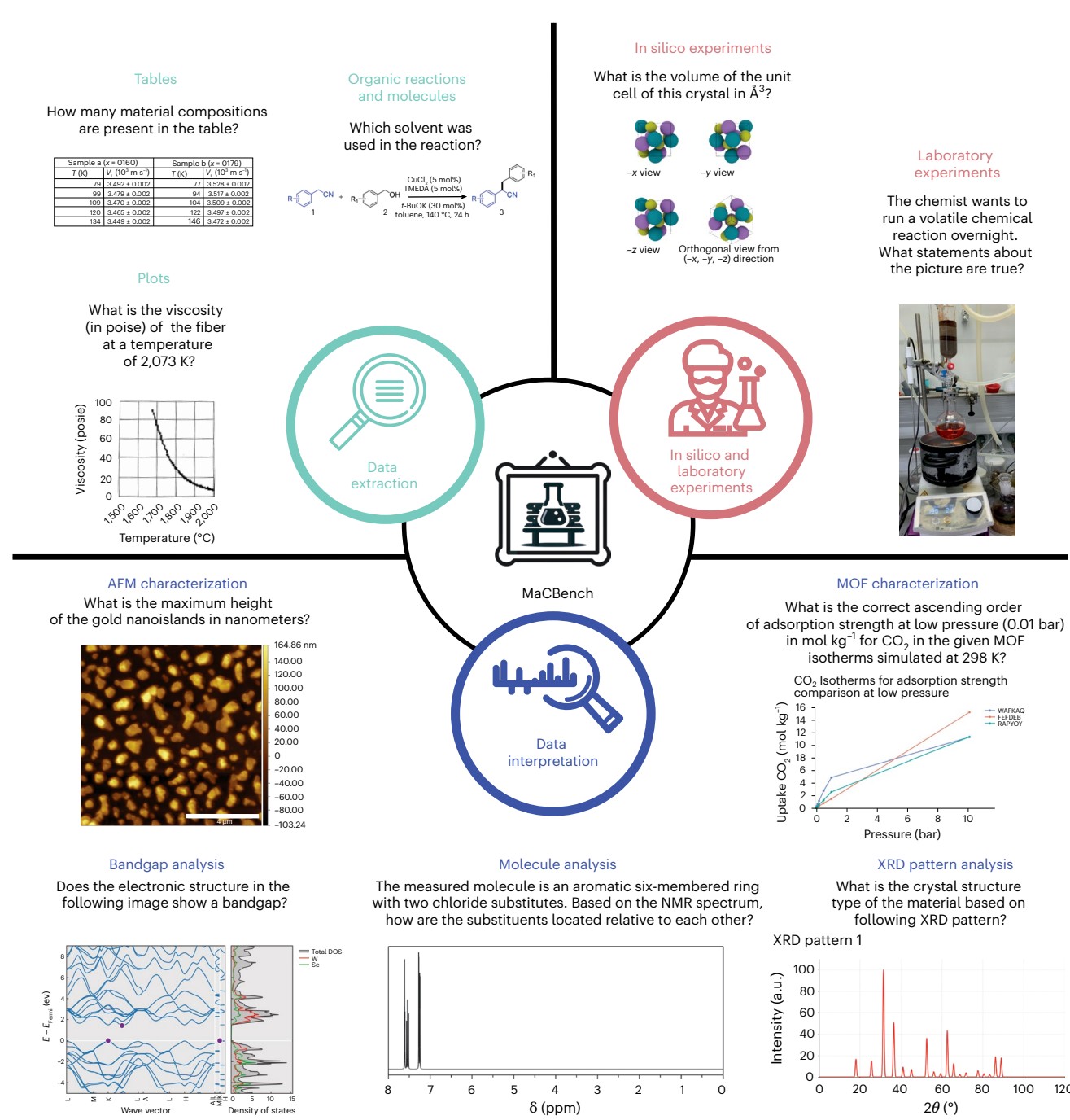

**Fig. 1 | Overview of the MaCBench framework covering the multimodal chemistry and materials science research life cycle.** The framework evaluates VLLM performance across three key focus areas: data extraction (teal), in silico and laboratory experiments (purple) and data interpretation (blue). The benchmark includes diverse tasks spanning tables, plots, organic chemistry diagrams, crystal structures, atomic force microscopy (AFM) imaging, spectroscopy and materials characterization. Each task requires domain-specific visual understanding and scientific reasoning, from extracting numerical values to analyzing complex experimental set-ups and interpreting spectroscopic data. Credit: icons, Rainy Ting, svgrepo.com.

is indistinguishable from random guessing in some cases, for example, in the assignment of space groups (see Supplementary Fig. 3).

**Data interpretation.** Interpreting experimental results often proves challenging for all models, including Claude 3.5 Sonnet. Although most models can interpret capacity values (average accuracy of 0.59), compare Henry constants from metal–organic framework isotherms (average accuracy of 0.83), or interpret amorphous or crystalline systems from XRD with acceptable performance (average accuracy of 0.69), they struggle to interpret AFM images (average accuracy of 0.24) and often fail with tasks that involve measurements such as width and length (despite the presence of clear legends). They also fail to reliably interpret mass spectrometry and nuclear magnetic resonance spectra (average accuracy of 0.35), or to make inferences on the XRD pattern. In the XRD case, it is particularly striking that although some models perform very well at identifying the positions of the most intense reflections, they perform poorly in determining relative orderings, which is crucial for interpreting XRD patterns.

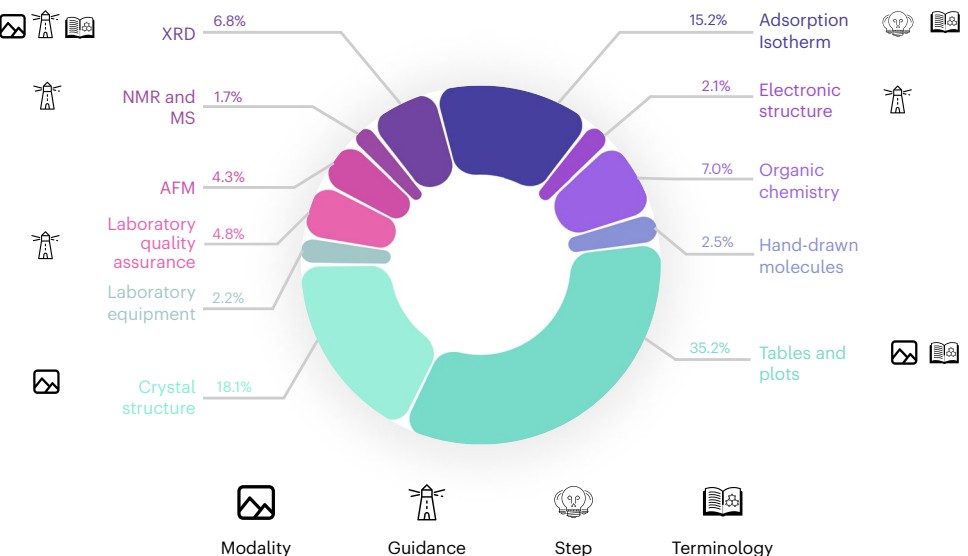

**Fig. 2 | Distribution of tasks in the MaCBench dataset.** MaCBench comprises eleven distinct topics with their respective proportions, ranging from tables and plots (35.2%) to mass spectrometry (MS) and nuclear magnetic resonance (NMR) analysis (1.7%). Each segment is annotated with relevant icons indicating the ablations we conducted on those tasks: modality understanding (image icon), guidance requirements (lighthouse icon), reasoning steps (lightbulb icon) and terminology complexity (book icon). The chart illustrates the benchmark's comprehensive coverage of chemistry and materials tasks.

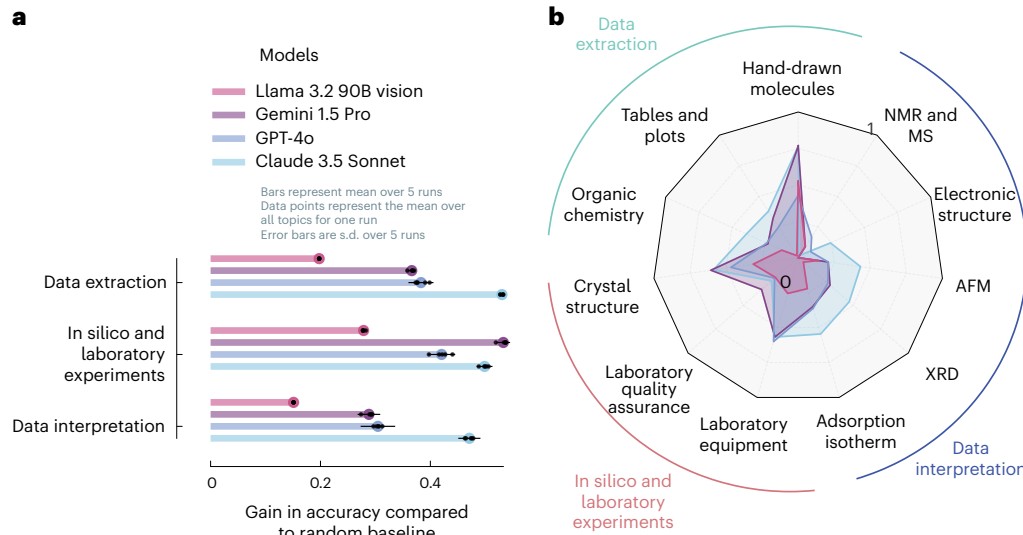

**Fig. 3 | Performance of frontier VLLMs. a**, Accuracy gains versus random baselines across three focus areas, showing the varying performance of Claude 3.5 Sonnet, GPT-4o, Gemini 1.5 Pro and Llama 3.2 90B Vision when averaged across all tasks in the three MaCBench focus areas: data extraction, experimental understanding and interpretation. We show performance as a fraction of correctly answered questions relative to a random baseline. A performance of 0 means that the model is indistinguishable from random guessing. The error bars indicate the s.d. of the fraction of correctly answered questions over five different runs. **b**, Radar plot demonstrating the relative model performance across topics. Again, we show the fraction of correctly answered questions relative to a random baseline (the plots without the normalization are shown in Supplementary Fig. 2).

## Understanding model limitations

We designed a comprehensive suite of ablation studies to further understand the failure modes of VLLMs. Our approach isolates specific aspects of scientific tasks, from the complexity of the reasoning required, to how the information is presented. We probe two distinct categories of limitations (Fig. 4): first, core reasoning limitations that seem fundamental to current model architectures or training approaches or datasets, and second, sensitivities to inference choices.

**Core reasoning limitations.** Some limitations seem to be intrinsic to current model architectures and are unlikely to be overcome regardless of how tasks are presented or prompted. These fundamental constraints manifest in three key areas.

*Spatial reasoning.* Although one might expect VLLMs to excel at processing spatial information, our results reveal substantial limitations in this capability. For example, although models achieve high performance in matching hand-drawn molecules to simplified molecular input line-entry system (SMILES) strings (average accuracy of 0.80, four-times better than baseline), they perform almost indistinguishably from random guessing at naming the isomeric relationship between two compounds (for example, enantiomer, regioisomer, average accuracy of 0.24, which is only 0.1 higher than the baseline accuracy) and

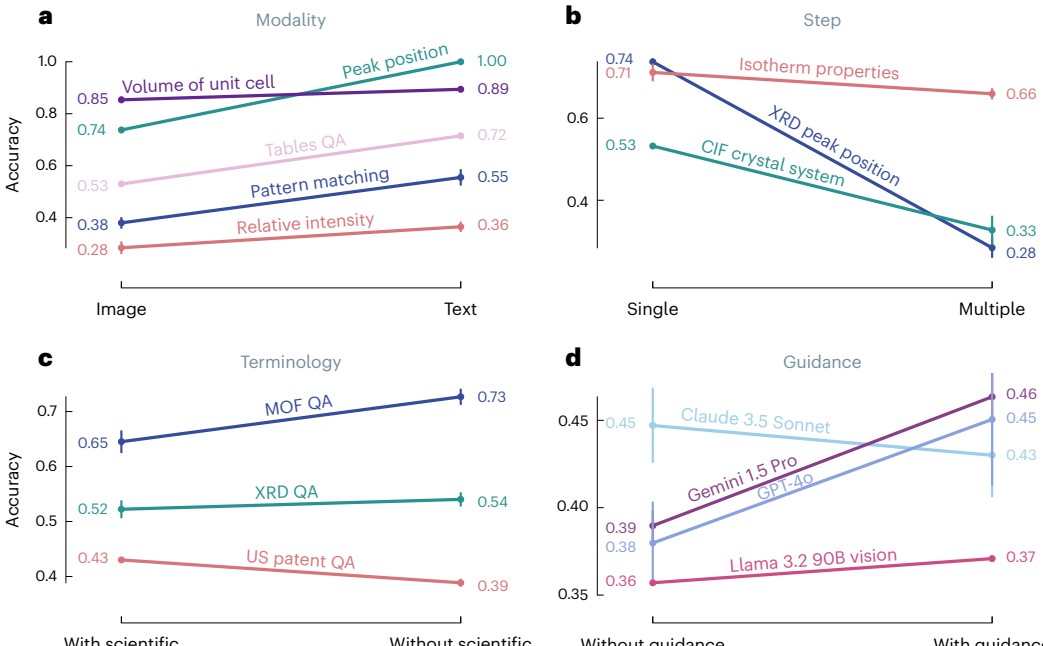

**Fig. 4 | Ablation study results across four key dimensions of VLLMs performance in chemistry and materials science tasks. a**, Modality analysis compares performance between image- and text-only inputs across different task types, with typically higher performance when the same information is shown in text form. **b**, Step complexity analysis demonstrates performance degradation as tasks require multiple reasoning steps. **c**, Terminology impact shows how scientific language specificity affects model accuracy, comparing performance with and without domain-specific terminology. We found the behavior on US patent quality assurance (QA) to be mostly due to the sensitivity of Gemini 1.5 Pro to the prompt template (see Supplementary Section 6). **d**, The guidance study compares performance across different VLLMs with and without additional task guidance, revealing model-specific sensitivity to prompting strategies. For

each task, we calculated the mean score and s.d. across five independent runs. Points and error bars represent the mean and s.d. over five independent runs. respectively. We averaged the mean scores and s.d. for each task to summarize performance across models. We employed a two-step averaging process for topics (for example, 'XRD QA', 'Isotherm QA', 'Tables QA'). We averaged the scores and s.d. across the tasks for each model. We then averaged these model-specific averages across all models to obtain the final mean score and s.d. for the topic. For guidance analysis, performance was measured as the mean score across five independent runs, and the variability was quantified using the s.d. of those runs. To obtain an overall measure of performance and variability for each side (with and without guidance), we calculated the mean score and the mean s.d. across all tasks within each side.

when assigning stereochemistry (average accuracy of 0.24, baseline of 0.22). Similarly, models perform well in simple perception tasks on crystal structures (for instance, when counting the number of different species, average accuracy of 0.85) but struggle at assigning the crystal system (average accuracy of 0.55) or space groups (average accuracy of 0.45).

These performance drops for tasks requiring spatial reasoning suggest that current VLLMs cannot reliably be used for any tasks requiring this capability—even though this might be one of the most intuitive use cases of these models.

*Synthesis across modalities.* Given that models consume visual and textual input in seemingly similar ways, one might expect that the same information is processed in the same way regardless of how it is presented to the model.

We presented identical text and image information to probe the ability of models to integrate information across modalities. In Fig. 4 we find that for all tasks in which we show the same information, the performance in the text modality is better than when the information is provided as an image. A striking example emerges when identifying the peak position in XRD. Models show a nearly 35% increase in performance when presented with the same peak positions as text versus showing the peaks visually. Even when calculating the volume of crystal structures, models differ in performance by four percentage points when presented with the structural information in visual (unit cell parameters shown in the image) and textual (unit cell parameters shown in text) forms. These results suggest that current models have not yet developed robust strategies for cross-modal information synthesis.

*Multi-step reasoning.* Motivated by the fact that the overall performance analysis indicated that perception tasks tended to perform best, we designed experiments in which we probe—with the same inputs[46]—performance on very similar tasks, but with varying numbers of reasoning steps (or different numbers of tool calls when implemented in an agentic framework).

Our analysis reveals consistent degradation in performance as tasks require more reasoning steps. Figure 4 shows that in all of our experiments, the tasks that require multiple steps perform substantially worse than those requiring only one. For instance, in XRD pattern analysis, models perform much better at identifying the highest peak than at ranking relative peak intensities (average accuracy of 0.74 for identification of the highest peak versus 0.28 for ranking). Similarly, for the interpretation of adsorption isotherms, the accuracy in finding the highest value notably exceeds the performance of ordering multiple values. This pattern suggests fundamental limitations in chaining logical steps—a crucial capability for scientific reasoning.

**Sensitivity to inference choices.** Although addressing these core limitations will require novel training approaches, we also identified several factors that substantially influence model performance through inference choices rather than fundamental capabilities. Those factors present an actionable way to improve the performance of current systems directly without retraining them.

*Scientific terminology.* One might hypothesize that models struggle with some tasks because they are unfamiliar with the scientific terminology used in the questions. Figure 4 shows that removing scientific

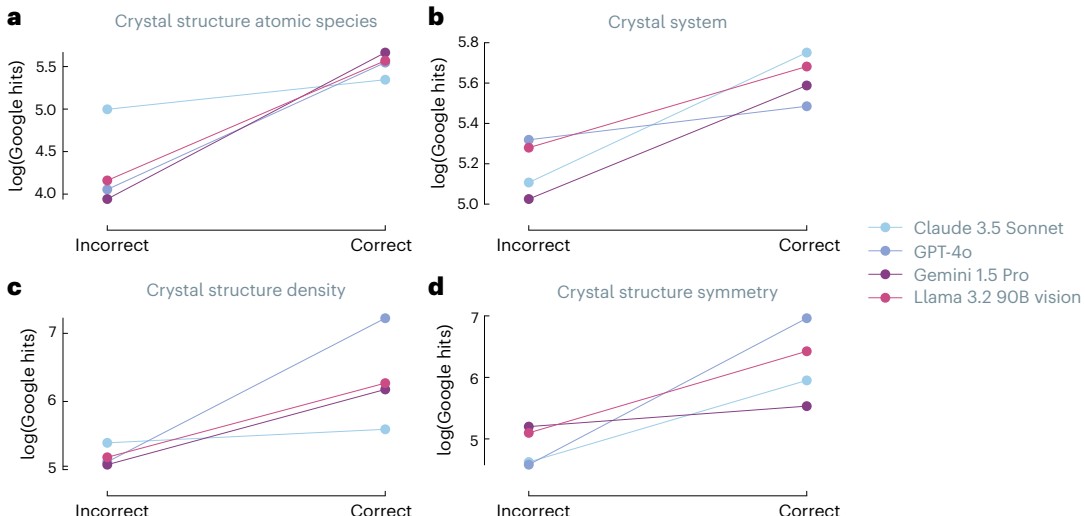

**Fig. 5 | VLLM performance as a function of number of search hits.**
**a–d**, The plots compare four leading VLLMs across different crystallographic tasks: atomic species identification (**a**), crystal system classification (**b**), crystal structure density calculations (**c**) and crystal symmetry determination (**d**). For each property, log-scale Google hit counts are plotted against the binary correctness (correct/incorrect) of model responses, with lines serving as visual aids only, revealing correlations between answer accuracy and the prevalence of information in online sources. Higher hit counts for correct answers suggest models may not solely rely on reasoning in their responses to crystal structure analysis tasks.

terminology improves performance across some tasks, including the analysis of adsorption isotherms of metal–organic frameworks, XRD pattern interpretation. Similarly, using International Union of Pure and Applied Chemistry (IUPAC) names instead of SMILES notation for chemical compound identification leads to better results. This suggests models might be overly sensitive to specific technical vocabularies rather than understanding underlying concepts. In fact, some models such as Gemini 1.5 Pro (and the surrounding refusal mechanisms) are very sensitive to the exact wording of the prompt. In Supplementary Section 6, we show that for some questions, large variations in performance can be due to apparently minor changes in prompt wording, such as replacing the word 'image' with 'diagram,' 'plot,' 'figure,' 'photograph', or even omitting it entirely.

*Guidance following.* Given that chemists receive instructions on interpreting various experimental characterizations, we hypothesized that similar guidance might also help the models perform better on such tasks. Interestingly, adding step-by-step instructions improves performance for most models in spectral analysis, electronic structure interpretation and XRD pattern matching—with the notable exception of Claude 3.5 Sonnet, whose performance does not improve when provided with guidance. This variation in response to instruction suggests different underlying approaches to problem solving across models.

### Performance as a function of frequency on the Internet

The varying impact of guidance across models led us to investigate whether models truly engage in scientific reasoning or primarily match patterns from their training data[46]. To probe this question, we measured the number of Google search results for various crystal structures as a proxy for the frequency of those structures in the training corpus (Fig. 5).

Our analysis reveals a correlation between the prominence of crystal structures on the Internet and task performance. Figure 5 shows that for all cases in our benchmark, the structures for which the models solve the tasks are more prominent on the Internet. This suggests that models might rely more on pattern matching than genuine scientific reasoning. Interestingly, we observe this effect even for tasks that depend solely on perception, such as counting the number of distinct atomic species.

### Toward robust multimodal assistants

Our analysis reveals the promise and limitations of state-of-the-art VLLMs in scientific tasks. Compared with text-only benchmarks such as that of Mirza and colleagues[40], we observe much higher performance variability across tasks, suggesting that multimodal systems are more fragile than LLMs. This fragility manifests in several ways: the striking performance gap between visual and textual representations of identical information indicates incomplete integration of modalities, whereas the strong correlation between model performance and the Internet presence of specific crystal structures raises questions about true reasoning capabilities versus pattern matching. The sensitivity to prompting choices (see Supplementary Section 6) and the counterintuitive finding that guidance can degrade performance for top models further underscore reliability concerns; however, our findings also point to actionable paths forward. Many observed limitations, particularly in spatial reasoning, could be addressed through synthetic training data generation. When pursuing such approaches, we recommend incorporating generalization tests (for example, evaluating spatial reasoning on larger compounds than those in training[47]) to ensure robust capability development. Furthermore, the substantial performance differences between modalities suggest opportunities for improved training strategies, such as incorporating modality transformation tasks (for example, automated conversion between spectral data representations). These targeted interventions could help bridge the gap between current capabilities and the needs of scientific workflows. Looking forward, it is also important to note that for future workflows, with advanced data management[48] or self-driving laboratories[49], some of the tested multimodal integration abilities will be less important as data will directly be available in a machine-actionable form instead of requiring parsing from an image.

### Discussion

Scientific reasoning is fundamentally a multimodal process. Current vision language models show promising capabilities in simple cases, such as identifying laboratory equipment or extracting explicit numerical values from plots. For standardized representations such as SMILES notations or simple spectra, models can even achieve high accuracy in information extraction; however, model performance becomes unreliable when tasks require the integration of visual and conceptual understanding—as in complex laboratory safety assessments or crystal structure analysis.

Through careful ablation studies, we found that despite their impressive scale and training, current VLLMs require a substantial improvement in their vision modality as they seem to perform drastically better when the same information is shown as text instead of as an image. Moreover, the models seem to rely on pattern matching rather than developing robust scientific understanding. This becomes particularly evident in the observation that model performance correlates strongly with online prominence.

Yet our benchmark also demonstrates the remarkable progress in AI systems' ability to process scientific information, with (almost) perfect performance achieved in several tasks. The observation that performance can be improved through careful terminology choice and task guidance (albeit with model-specific variations) suggests practical paths forward. More broadly, our findings indicate that advancing AI in science requires not just model improvements but also better ways of representing scientific knowledge—particularly when addressing the observed gaps in spatial reasoning and cross-modal integration capabilities.

Although current VLLMs cannot yet serve as autonomous scientific reasoners, they show promise as assistive tools when their limitations are well understood and their deployment is carefully structured around their demonstrated strengths. Although our diverse benchmark offers insights, it does not encompass the full scope of scientific reasoning, and the LLMs evaluated—although representative—are not exhaustive of all available architectures. These limitations highlight the need for continued research across a wider array of tasks and models to comprehensively evaluate AI's capabilities and shortcomings in science. Furthermore, interpretability studies are crucial for understanding the reasoning behind these models' outputs, ensuring that progress signifies genuine scientific comprehension, not just high performance.

As we continue to develop these systems, our work suggests that advancing from pattern matching—demonstrated by the strong correlation between model performance and internet presence of crystal structures—to true scientific reasoning may require fundamental advances in both training data curation and model architectures that can better handle spatial relationships and cross-modal information synthesis.

## Methods

Our question curation and model evaluation methodology leverages the ChemBench framework[8]. For curation, we manually sourced questions and then created ablations based on error analyses to systematically understand failure modes (an illustration of curation workflow is shown in Supplementary Fig. 1). We created new images for most tasks, for example, by building and photographing laboratory set-ups or by plotting experimental data. Similar to Mirza and colleagues[40], all questions have been reviewed by multiple scientists before being entered into the corpus. In the curation process, we also recorded tolerances for each question; that is, for each numerical answer, we recorded windows within which an answer would still be deemed correct to account for natural uncertainties and noise.

### Dataset

Our questions in the dataset are stored in an extended BigBench format[50]. Each question, along with its corresponding base-64-encoded image, is stored in separate JSON files. The BigBench canary string is included in each file to prevent potential data leakage during future model training. Our pipeline employs a robust templating system, allowing for the dynamic insertion of multiple images and other text template elements into questions using placeholders. This enables our pipeline to interleave images directly into question prompts in designated locations.

All questions in our benchmark contain pairs of images and text-based questions. Only some ablation experiments (which are specifically highlighted) contain only text information.

### Evaluation

We employ Mirza and co-workers' prompt templates and parsing workflow[40], which uses regex-based functions to extract answers from various scientific notations, handling both MCQ responses and numerical values. The regex-based parsing is backed up with an LLM extractor (for example, Claude 3.5 Sonnet) for cases in which standard parsing fails. We included the encoded images in the prompt. We used the default quality setting for each provider. That is, for Gemini 1.5 Pro, images will be automatically scaled up or down to fit into the allowed range (768 × 768 – 3072 × 3072), whereas for Claude 3.5 Sonnet, the image's long edge is scaled down if it is greater than 1,568 pixels. For Llama 3.2 90B Vision, an application programming interface (API) error will be raised if the images are too large. For GPT-4o, the default configuration is set to 'auto', meaning that the quality of the images is automatically selected by the API. Low-resolution images are set to 512 × 512 pixels. For the high-resolution mode, the model first sees the 512 × 512 image and then the image is cropped into tiles that are studied individually.

### Scoring methodology

For MCQs, a task is considered correct if the Hamming loss is zero, meaning the predicted answer exactly matches the ground truth. For numeric questions, a response was deemed correct if the mean absolute error falls within a specified tolerance. The default tolerance is 1%, but the tolerance can reach up to 5% for certain question types, such as CIF-density, CIF-volume (where CIF is crystallographic information file) and some US patent questions (the tolerance is defined in the curation process). Each correct task receives a score of 1, whereas incorrect tasks receive a score of 0.

### Overall and topic scores

The overall score is calculated as the total number of correct tasks divided by the total number of questions, excluding ablation tasks. Topic-wise scores are computed similarly, with the total number of correct answers in a topic divided by the total number of questions in that topic. The same topic-wise scoring method is applied for ablation tasks. When tasks or models are combined, their respective scores and the s.d. are averaged.

### Baseline

The random baseline is established by randomly selecting one answer from the available options for MCQs. For example, if there's a MCQs chemistry question asking 'Which element has the highest electronegativity?', with options (a) fluorine, (b) oxygen, (c) nitrogen and (d) chlorine, the baseline would simply pick one letter randomly (for example, 'c'). For numeric questions, we use the mean of all target values within a topic as the prediction. For instance, if there are multiple questions asking for the number of atoms in CIFs with answers such as 6, 12 and 15, the baseline would calculate the average of all these values (11) and use that as its prediction for every numeric question in that topic. The entire benchmark was run five times, and the s.d. of the overall and topic-wise scores is used as the error bar to account for variability.

### Refusal

We implement a comprehensive framework combining regular expression-based detection from LLM Guard and a fine-tuned BERT model[51] to identify potential LLM refusals. This detection pipeline was integrated into our evaluation pipeline, enabling pre-scoring refusal checks. To mitigate refusals, we implemented an interval-based retry mechanism, requerying the LLM up to $n$ times until a non-refusal response was obtained. For our runs we retried a maximum of five times. A count on the refusal by different models is shown in Supplementary Table 5.

### Relative performance

To account for the fact that a non-zero performance can be achieved for MCQs via random guessing, which is also dependent on the number

of options, we report the metrics in the main text as performance gains relative to what the random baseline would achieve:

$$acc_{rel} = acc - acc_{baseline} \qquad (1)$$

### Correlation of performance with the number of search results

To analyze the correlation between the performance of the models and the prominence of the web, we used the total number of results for querying the common name of crystal structures returned by the Serp API.

### Evaluation card

We describe the benchmark design and governance in an evaluation card[52] that is accessible via GitHub at https://github.com/lamalab-org/macbench/blob/main/eval-card.md (ref. 53).

## Data availability

To facilitate the benchmarking and reproducibility of our work, we have provided the datasets (v.1.0.0 for the MaCBench dataset) used in this work on HuggingFace[54,55]. Source Data are provided with this paper.

## Code availability

The code for running the benchmark is available at https://github.com/lamalab-org/chembench/ (ref. 56) and archived on Zenodo[57]. We used v.0.3.0 for this study. Instructions on how to run the benchmark can be found at https://lamalab-org.github.io/chembench/getting_started/#how-to-benchmark-on-multi-modal-tasks.

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

## Acknowledgements

K.M.J. acknowledges the support by the Carl Zeiss Foundation, and a 'Talent Fund' of the 'Life' profile line of the Friedrich Schiller University Jena. A grant from OpenPhilanthropy further supported parts of the work. Furthermore, M.S-W. was supported by Intel and Merck via the AWASES program. K.M.J. is part of the NFDI consortium FAIRmat funded by the Deutsche Forschungsgemeinschaft (German Research Foundation; project no. 460197019). N.M.A.K. acknowledges the Google Research Scholar Award, the Alexander von Humboldt Foundation for funding support, and the HPC IIT Delhi for computational and storage resources. We thank B. Rieck for developing the LaTeX-credit package (https://github.com/Pseudomanifold/latex-credits; ref. 58). We also thank K. Schreyer for helping collect the pictures for the laboratory quality assurance task, N. N. Gosvami for providing the AFM images and M. Zaki for support with table and figure data.

## Author contributions

N.A. and M.R.G. developed the software for the benchmarking framework. N.A. M.S.W. and M.R.G. created the visualizations for figures and tables. All authors contributed to the project's conceptualization, data curation, investigation, methodology and validation of results. K.M.J. authored the original manuscript draft. All authors participated in the review and editing process. K.M.J. and N.M.A.K. managed funding acquisition, project administration, resources, and provided supervision throughout the project.

## Funding

## Competing interests

K.M.J. has been a paid contractor for OpenAI (as part of the red teaming network). The remaining authors declare no competing interests.

## Additional information

**Correspondence and requests for materials** should be addressed to N. M. Anoop Krishnan or Kevin Maik Jablonka.

