## [Peer Review file · Nature Computational Science]

Probing the limitations of multimodal language models for chemistry and materials research

Corresponding Author: Dr Kevin Maik Jablonka

Version 0:

Decision Letter:

** Please ensure you delete the link to your author homepage in this e-mail if you wish to forward it to your co-authors. **

Dear Dr Jablonka,

Your manuscript "Probing the limitations of multimodal language models for chemistry and materials research" has now been seen by 3 referees, whose comments are appended below. You will see that while they find your work of interest, they have raised points that need to be addressed before we can make a decision on publication.

The referees' reports seem to be quite clear. Naturally, we will need you to address **all** of the points raised.

While we ask you to address all of the points raised, the following points need to be substantially worked on:

- 1) Please improve the figures based on the recommendations of the reviewers
- 2) Please ensure that your code repository provides all necessary data, instructions and documentations to ensure reproducibility.
- 3) Please demonstrate the reproducibility of the experiments/benchmarking

Please use the following link to submit your revised manuscript and a point-by-point response to the referees' comments (which should be in a separate document to any cover letter):

Link Redacted

** This url links to your confidential homepage and associated information about manuscripts you may have submitted or be reviewing for us. If you wish to forward this e-mail to co-authors, please delete this link to your homepage first. **

To aid in the review process, we would appreciate it if you could also provide a copy of your manuscript files that indicates your revisions by making use of Track Changes or similar mark-up tools. Please also ensure that all correspondence is marked with your Nature Computational Science reference number in the subject line.

In addition, please make sure to upload a Word Document or LaTeX version of your text, to assist us in the editorial stage.

To improve transparency in authorship, we request that all authors identified as 'corresponding author' on published papers create and link their Open Researcher and Contributor Identifier (ORCID) with their account on the Manuscript Tracking System (MTS), prior to acceptance. ORCID helps the scientific community achieve unambiguous attribution of all scholarly contributions. You can create and link your ORCID from the home page of the MTS by clicking on 'Modify my Springer Nature account'. For more information please visit please visit www.springernature.com/orcid.

We hope to receive your revised paper within three weeks. If you cannot send it within this time, please let us know.

Best regards,

Kaitlin McCardle, PhD
Senior Editor
Nature Computational Science

Reviewers comments:

Reviewer #1 (Remarks to the Author):

Review for Probing the limitations of multimodal language models for chemistry and materials research

This article is a well written assessment of the laboratory assistant AI landscape in the form of a benchmarking package. The authors organize tasks into three groups, each containing sets of questions and tasks pertaining to experimentation, data interpretation, and data extraction. There are a lot of clever experiments gauging each model's ability to reason and perform in a variety of tasks. Overall excellent work and I recommend this manuscript for publication.

Some comments:

1. Page 1- Fig 1- In the text the three categories of questions are described as pillars, which I thought was apt. Why is it a cycle in Figure 1?
2. Page 6- Fig 3B- I think the radar plot is a bit hard to interpret, but I think it is a nice-looking figure.
3. On page 7, "limitations appear intrinsic to current model architectures". Why is it not possibly just a lack of training data?
4. I would be curious about a 3-D vs 2-D split, or different r
5. I think including safety as its own desired property in the table A.1 would be suitable
6. I was unable to install the package but all of the questions are there which is the major contribution of the codebase.

Reviewer #1 (Remarks on code availability):

Following the instructions to install did not work as some internal functions are no longer working and crashing the program. Cloning the repository and installing it didn't fix the problem either. I tried to troubleshoot a bit but ultimately decided to leave it. The dataset is all there.

Reviewer #2 (Remarks to the Author):

In their manuscript "Probing the limitations of multimodal language models for chemistry and materials research", Alampara et al. describe a benchmark for evaluation of large language models' (LLMs) capability of analyzing chemical information. The authors subject four well-known LLMs to their benchmark, compare their performance and draw conclusions from their responses. I very much welcome the authors' timely publication, scrutinizing the hype around LLMs and their employment in every possible field of science. I found the article very interesting, and answering a lot of burning questions around the possibilities of LLMs. The benchmark, published for free and open access, has the potential to become a very useful testing tool for LLMs, at least until LLMs become specifically trained on it. I have some questions, comments and suggestions about the reported research and the manuscript itself, but I would strongly recommend the manuscript for publication in this journal.

Can the authors comment about the reproducibility of their benchmark results? As I do not see statistical analysis of any of the responses, I assume each question was asked just once. I find it interesting whether repeating the same question for the same LLM yields different results. If so, deviations between them are as well quite interesting.

I find the benchmark baseline somewhat under-described. The authors often mention the "random baseline", but it is not clear to me how it works. For instance, Table A.2 shows the baseline for the extraction of organic reactions schema as 50%. I assume this means randomly extracted schemas are correct 50% of times, but to me such ratio seems very high. I cannot think of a program which randomly selects correct solvents, temperature and yield from organic reaction schemas half of the time. Maybe if I have the reaction split by components and randomly select one of them, I can reach 50% correct identification for the solvent, but some piece of software would have to do the reaction splitting. Moreover, this holds true only if 50% of components in reactions are solvents. It would be great if the authors could clarify all the "random baseline" selection methods. Finally, I find it really surprising that Llama 3.2 90B Vision fails answering all the questions in "Organic Reactions Schema" category, but this might be due to prompt refusals.

It would be really interesting to see the established pre-AI tools used as a baseline. First, it would make it much easier to understand the concept of the baseline (as indicated above). Second, it would make it much clearer to see how the LLMs rank when compared to the most advanced pre-AI open software tools, such as OPSIN, OSCAR4, OSRA and ChemicalTagger. I understand this suggestion is out of scope for the current manuscript, thus I do not expect it to be answered in the revised manuscript.

The authors refer to the tested LLMs as VLLMs (Vision Large Language Models). Why this emphasis of "visual" is needed, when the models described in the paper are also subjected to text-only prompts? It might as well be that I am missing the distinction as I am not an expert in the AI field.

When looking at Table A.1, it is not clear to me which of the questions are text-only, image-only and which are text+image. I find this distinction difficult to navigate in the remaining paper as well.

Table A.2 uses bold script to indicate "winners" of each challenge. I find this slightly misrepresenting, as 0.93 gets dominated by 0.97, eclipsing the former and putting it on the same shelf as 0.24. I am not entirely sure how to represent this better, but maybe color gradient could be used, making the best value of maximum color saturation and others at relative? However, authors may disregard this comment altogether, as I do not have a definite solution, and my suggestion might make the table even more complex.

Methodology for calculating the overall performance in Table A.2 needs to be explicit. Is it just the average from all the rows in the table?

Part of subsection "Scientific terminology" seems missing. The last sentence of this subsection ends abruptly without a period.

I have some suggestions how to make the figures easier to read. In Figure 3 part A, there are different ordering of models given in model enumeration and comparison to baseline (i.e., in the upper part cyan color is shown on top, and in the bottom part on bottom). I would suggest retaining the same order everywhere. Moreover, I personally find it difficult to distinguish the colors used for Gemini and Llama models. I understand the visual appeal of the current pastel palette, but please consider using more distant colors. In Figure 3 part B, the same model colors (or very similar to them) are used for drawing group arcs in "Data extraction", "Data interpretation" and so on. This might lead to thinking that these groups are somehow related to particular same-colored LLMs. Please use different colors for unrelated things. Same comments hold true for Figure A.1.

Please make a stable release of your benchmark and refer to this version in the paper. Since it is hosted on GitHub, I would suggest making a git tag, possibly v0.1.0 or v1.0.0.

Figures A.2 and A.3 could use the color legend. I find it tiring needing to jump through the pages to look at the legend in Figure A.1.

Table A.5 is very interesting and greatly supplements Table A.2, however, I personally find it difficult having to jump from one to another. I would suggest the authors consider merging Table A.5's information in Table A.2, for example by giving the number of rejected questions in brackets next to their correctness value. I am not entirely sure this will not overcrowd the Table A.2, thus I leave the decision to the authors.

I have some requests for the leaderboard. First, it lacks a clear link to the used benchmark, without it, it is difficult to understand the testing methodology for a passer-by. Next, please use a stable version of the benchmark for the leaderboard. This benchmark version should be clearly indicated in the leaderboard. Finally, please consider versioning the leaderboard itself and making it easier to compare between different versions of the leaderboard. This will become very important with gradual growth of the benchmark (if foreseen), addition, updates and possibly removals of LLMs from the evaluation set.

Some minor remarks:

In the caption for Figure 3, "across all task" should probably be written as "across all tasks" (plural).

Abbreviation "AFM" is first introduced in Figure 1, but I feel it should as well be un-abbreviated on its first occurrence in the text, in subsection "Data interpretation".

In "Data interpretation", "reflexes" should probably be written as "reflections" (seems to be the preferred term in XRD).

In the caption for Figure 4, I find the enumeration of VLLMs "(Claude 3.5 Sonnet, ...)" superfluous, as this is evident from the legend.

First sentence of "Synthesis across modalities", "Given that models input visual and textual input ...", I find the multiple usage of word "input" as different parts of speech tricky to read. In the same subsection "four percentage" could probably be better written as "4%", seemingly consistent with the style of the manuscript.

In Figure 6, "Ablation studys" should probably be written as "Ablation studies".

In "A.2 Related work", "Khalighinejad et al. build" should probably be written as "Khalighinejad et al. built" (past tense, consistent within the section).

Reviewer #2 (Remarks on code availability):

I did not have the possibility to have an in-depth review of the code, but I find it important to note that the code is MIT-licensed. The code needs a stable release, though. I asked the authors to make it in my review text as well.

Reviewer #3 (Remarks to the Author):

In the manuscript "Probing the Limitations of Multimodal Language Models for Chemistry and Materials Research," the authors make an effort to explore the application of large language models (LLMs) in the field of chemistry. However, I have several concerns regarding the current state of the manuscript, which I believe need to be addressed before considering it for publication.

One of the primary concerns is the reproducibility of the results presented in the manuscript, particularly regarding the use of context windows, which raises questions about the reliability of the findings. The absence of a system prompt to generate informed responses is a significant oversight. Including a system prompt could enhance the model's ability to generate contextually relevant and accurate outputs, thereby improving the reproducibility of the experiments.

The manuscript discusses the generation of images but fails to address the general applicability of these images. It is unclear why the authors did not utilize images from published sources, which could have provided a benchmark for comparison. The inclusion of such images would strengthen the validity of the results and offer a clearer understanding of the model's capabilities.

The definition of the random baseline is ambiguous, particularly concerning the random component. A clear explanation of how the random baseline is established and its role in the benchmarking process is essential for readers to assess the validity of the comparisons made in the study.

The manuscript lacks a discussion on strategies to alleviate the limitations of multimodal LLMs in the context of chemistry tasks. Additionally, the potential benefits of employing modern techniques, such as retrieval-augmented generation, are not explored. These techniques could significantly enhance the model's performance in retrieving accurate data and should be considered in future iterations of the study.

Overall, while the manuscript presents an interesting exploration of LLMs in chemistry, the concerns outlined above hinder its suitability for publication in its current form. I recommend that the authors address these issues to improve the clarity, reproducibility, and overall quality of the work. I would not recommend the publication of this manuscript in its current form.

Version 1:

Decision Letter:

**** Please ensure you delete the link to your author homepage in this e-mail if you wish to forward it to your co-authors. ****

Dear Dr Jablonka,

Your manuscript "Probing the limitations of multimodal language models for chemistry and materials research" has now been seen by 3 referees, whose comments are appended below. You will see that while they find your work of interest, they have raised points that need to be addressed before we can make a decision on publication.

The referees' reports seem to be quite clear. Naturally, we will need you to address ***all*** of the points raised.

While we ask you to address all of the points raised, the following points need to be substantially worked on:

- 1) Please improve the figure quality, as requested by the reviewers
- 2) Please provide clarifications on the questions list, as requested by Reviewer #2

Please use the following link to submit your revised manuscript and a point-by-point response to the referees' comments (which should be in a separate document to any cover letter):

Link Redacted

**** This url links to your confidential homepage and associated information about manuscripts you may have submitted or be reviewing for us. If you wish to forward this e-mail to co-authors, please delete this link to your homepage first. ****

To aid in the review process, we would appreciate it if you could also provide a copy of your manuscript files that indicates your revisions by making use of Track Changes or similar mark-up tools. Please also ensure that all correspondence is marked with your Nature Computational Science reference number in the subject line.

In addition, please make sure to upload a Word Document or LaTeX version of your text, to assist us in the editorial stage.

To improve transparency in authorship, we request that all authors identified as 'corresponding author' on published papers create and link their Open Researcher and Contributor Identifier (ORCID) with their account on the Manuscript Tracking System (MTS), prior to acceptance. ORCID helps the scientific community achieve unambiguous attribution of all scholarly

contributions. You can create and link your ORCID from the home page of the MTS by clicking on 'Modify my Springer Nature account'. For more information please visit www.springernature.com/orcid.

We hope to receive your revised paper within three weeks. If you cannot send it within this time, please let us know.

Best regards,

Kaitlin McCardle, PhD
Senior Editor
Nature Computational Science

Reviewers comments:

Reviewer #1 (Remarks to the Author):

Thank you for updating the manuscript, it is looking better! Here are some additional notes on this read. As well, the GitHub repository is a lot cleaner with improved documentation, which is appreciated.

1. Figure 1 images are grainy, if it's not just my version
2. Figure 3 fonts are non-standardized and not aligned
3. On page 6 data extraction is referred to as "the first step of the scientific process". This isn't right or wrong, but there's some back and forth between the sequential nature of the framework versus the three independent and equally important "pillars". If data extraction is the "first step", it might be worth considering explaining why the authors believe that, and how this affects model training.
 - o Considering that these models may be used as a part of larger agent-driven platforms, these semantics will be important to a reader trying to implement their own "AI scientist copilots". Implying a "starting point" may unintentionally bias a style of design, if it isn't relevant to the study at hand.
4. On a similar note, the terms "task", "task category", "core scientific tasks", "the three focus areas", "ten specialized scientific domains", "topics", "subtasks", etc. need to be standardized. I would suggest explicitly defining each tier and component of the benchmark in the introduction and using those terms consistently throughout the paper.
 - o Note a verb tense misalignment in the caption of figure 3A ("across all task")
 - o The error bars of figure 3A are barely visible
5. It may be worthwhile mentioning that some of the limitations are specific to VLLMs and not AI-driven experimentation (specifically as a preface in the Core Reasoning Limitations section). That is, it would be unlikely to directly have a VLLM interpret some experimental spectra from the image if the digital information was available and could be preprocessed.
 - o In the context of having a VLLM "read" (e.g., an android reading off a computer), such an evaluation makes sense.
 - o But in a cloud-based system where data is uploaded directly from an analytical instrument and digitally processed before digested by a central AI (which may be driven by a (V)LLM agent), its performance in identifying peaks directly from an image is less relevant.
 - o In general, it would be good to temper both positive and negative claims with the (in-lab) application context envisioned by the authors to better calibrate the reader's interpretations and take-aways
6. With respect to multistep reasoning, I think the audience would now be curious about the newer deepseek r1 and GRPO strategies, if the authors are able to incorporate.
 - o Admittedly, it's possible by the time the authors could, there might already be a new advancement in the field.
7. Figure 5 might say % correct in the x axis?
8. Was there any assessment performed on the model performance with respect to the original image quality (not the scaled resolution, but rather a study seeing how different amounts of noise in the source image can affect results?)
9. In terms of the desired properties enumerated in A1, there is a point about experimental planning. While it may be too broad of a scope to include as experiments in the current work, a discussion on model ability to directly interface with robotics may be appropriate.
 - o This was just published <https://pubs.acs.org/doi/10.1021/jacs.4c17738>
 - o I believe ORGANA has been published
 - o <https://www.nature.com/articles/s41467-024-54457-x>

Reviewer #1 (Remarks on code availability):

Working as described, but I would suggest adding the instructions of adding an API key to your environment within the repository readme .

Reviewer #2 (Remarks to the Author):

I would like to thank the authors for considering all my questions, suggestions and comments. The authors have greatly improved the manuscript, making it much easier to grasp the most important details of their study. However, I feel that couple of my original points have been answered only partially, thus I would like to request the authors for additional revision.

First, in the initial manuscript I completely overlooked the fact that the LLMs were solving the multiple-choice questions (MCQs). Only after the revision of the initial manuscript it came to my view that this was the case, in contrast to answering open-ended questions. It is still not clear to me whether all the questions were MCQs, or just part of them. I think this attribute of the study needs to be stressed even more, and has to be introduced as early as possible. Otherwise a reader might get an overly optimistic view of LLMs (like I did).

Correctness of LLMs' choices in MCQs vastly depends on the number of options, how close they are one to another and how well they sample the response space. Take a simple optical character recognition test for example: a recognizer is much more likely to choose "O" as correct answer if given "O", "I" and "X" to choose from than compared to "O", "0", "Q", "C", "D", or the whole alphabet. What is more, there is a need to know the correct answer as one of the possibilities a priori. Thus I expect a detailed discussion on how the options for MCQs were selected.

I am grateful for the authors to responding to my suggestions regarding versioning. Now ChemBench and all the data on the leaderboard is versioned. What I miss is the version of MacBench in its Hugging Face page. For MaCBench-Results I see "1.0.0" under "Split" tab under "Dataset Viewer", but the same does not hold for MacBench. GitHub page for MacBench does not have tags as well.

What is more, the manuscript should include the versions of all the software (I assume ChemBench) and all the data (MacBench) used to arrive to its conclusions. Please also add these versions to the text. I am aware that DOIs are provided, however, just by looking at DOIs (and even their landing pages) it will be difficult for the future reader to compare the versions (for example, to check whether major versions are the same etc).

Reviewer #3 (Remarks to the Author):

The new version has significant improvements with respect the first round for reviews. In my opinion, the authors addressed satisfactorily most if not all the reviewers concerns. I would recommend to publish this manuscript as is.

Version 2:

Decision Letter:

Our ref: NATCOMPUTSCI-24-2688B

15th April 2025

Dear Dr. Jablonka,

Thank you for submitting your revised manuscript "Probing the limitations of multimodal language models for chemistry and materials research" (NATCOMPUTSCI-24-2688B). It has now been seen by the original referees and their comments are below. The reviewers find that the paper has improved in revision, and therefore we'll be happy in principle to publish it in Nature Computational Science, pending minor revisions to satisfy the referees' final requests and to comply with our editorial and formatting guidelines.

TRANSPARENT PEER REVIEW

Nature Computational Science offers a transparent peer review option for original research manuscripts. We encourage increased transparency in peer review by publishing the reviewer comments, author rebuttal letters and editorial decision letters if the authors agree. Such peer review material is made available as a supplementary peer review file. **Please remember to choose, using the manuscript system, whether or not you want to participate in transparent peer review.**

Please note: we allow redactions to authors' rebuttal and reviewer comments in the interest of confidentiality. If you are concerned about the release of confidential data, please let us know specifically what information you would like to have removed. Please note that we cannot incorporate redactions for any other reasons. Reviewer names will be published in the peer review files if the reviewer signed the comments to authors, or if reviewers explicitly agree to release their name. For more information, please refer to our <https://www.nature.com/documents/nr-transparent-peer-review.pdf> target="new">FAQ page.

Thank you again for your interest in Nature Computational Science. Please do not hesitate to contact me if you have any questions.

Sincerely,

Kaitlin McCardle, PhD
Senior Editor
Nature Computational Science

ORCID

Reviewer #1 (Remarks to the Author):

We thank the author for their changes to the text and code to improve the clarity of the work

Reviewer #2 (Remarks to the Author):

I would like to thank the authors for considering all my comments. One small remark is that abbreviation "MCQ" is not explained on its first occurrence now, but on its second. However, this does not require additional revise-review cycle. I have no further comments.

Version 3:

Decision Letter:

Dear Dr Jablonka,

We are pleased to inform you that your Resource "Probing the limitations of multimodal language models for chemistry and materials research" has now been accepted for publication in Nature Computational Science.

Once your manuscript is typeset, you will receive an email with a link to choose the appropriate publishing options for your paper and our Author Services team will be in touch regarding any additional information that may be required.

Acceptance of your manuscript is conditional on all authors' agreement with our publication policies (see <https://www.nature.com/natcomputsci/for-authors>). In particular your manuscript must not be published elsewhere and there must be no announcement of the work to any media outlet until the publication date (the day on which it is uploaded onto our web site).

Before your manuscript is typeset, we will edit the text to ensure it is intelligible to our wide readership and conforms to house style. We look particularly carefully at the titles of all papers to ensure that they are relatively brief and understandable.

Once your manuscript is typeset, you will receive a link to your electronic proof via email with a request to make any corrections within 48 hours. If, when you receive your proof, you cannot meet this deadline, please inform us at rjsproduction@springernature.com immediately.

If you have queries at any point during the production process then please contact the production team at rjsproduction@springernature.com.

You may wish to make your media relations office aware of your accepted publication, in case they consider it appropriate to organize some internal or external publicity. Once your paper has been scheduled you will receive an email confirming the publication details. This is normally 3-4 working days in advance of publication. If you need additional notice of the date and time of publication, please let the production team know when you receive the proof of your article to ensure there is sufficient time to coordinate. Further information on our embargo policies can be found here:

<https://www.nature.com/authors/policies/embargo.html>

We welcome the submission of potential cover material (including a short caption of around 40 words) related to your manuscript; suggestions should be sent to Nature Computational Science as electronic files (the image should be 300 dpi at 210 x 297 mm in either TIFF or JPEG format). We also welcome suggestions for the Hero Image, which appears at the top of our [home page](http://www.nature.com/natcomputsci); these should be 72 dpi at 1400 x 400 pixels in JPEG format. Please note that such pictures should be selected more for their aesthetic appeal than for their scientific content, and that colour images work better than black and white or grayscale images. Please do not try to design a cover with the Nature Computational Science logo etc., and please do not submit composites of images related to your work. I am sure you will understand that we cannot make any promise as to whether any of your suggestions might be selected for the cover of the journal.

Best regards,

Kaitlin McCardle, PhD
Senior Editor
Nature Computational Science

P.S. Click on the following link if you would like to recommend Nature Computational Science to your librarian: <https://www.springernature.com/gp/librarians/recommend-to-your-library>

** Visit the Springer Nature Editorial and Publishing website at <http://editorial-jobs.springernature.com> for more information about our career opportunities. If you have any questions please click [here](mailto:editorial.publishing.jobs@springernature.com). **

RESPONSE TO THE REVIEWERS

REVIEWER 1

Reviewer Point P 1.1 — This article is a well written assessment of the laboratory assistant AI landscape in the form of a benchmarking package. The authors organize tasks into three groups, each containing sets of questions and tasks pertaining to experimentation, data interpretation, and data extraction. There are a lot of clever experiments gauging each model’s ability to reason and perform in a variety of tasks. Overall excellent work and I recommend this manuscript for publication.

Reply: *We thank the reviewer for taking the time to review our manuscript and are glad that you find our work valuable.*

Reviewer Point P 1.2 — 1. Page 1- Fig 1- In the text the three categories of questions are described as pillars, which I thought was apt. Why is it a cycle in Figure 1?

Reply: *We removed the arrows from the figure to clarify this aspect and to avoid misleading readers by potentially suggesting that research always follows a certain order.*

Reviewer Point P 1.3 — 2. Page 6- Fig 3B- I think the radar plot is a bit hard to interpret, but I think it is a nice-looking figure.

Reply: *To increase the readability, we increased the size of the radar plot.*

Reviewer Point P 1.4 — 3. On page 7, “limitations appear intrinsic to current model architectures”. Why is it not possibly just a lack of training data?

Reply: *We clarified this potentially misleading statement as follows.*

We probe two distinct categories of limitations (see Figure 4): first, core reasoning limitations that appear fundamental to current model architectures or training approaches or datasets, and second, sensitivities to inference choices.

Reviewer Point P 1.5 — 4. I would be curious about a 3-D vs 2-D split, or different r

Reply: *To address this point, we added a new ablation in which we compare the model performance in understanding the relationship between two isomers given 2D or 3D renderings of molecules.*

We describe the results in more detail in the appendix (see appendix section “Organic chemistry performance”).

We further observe that 3D-rendered molecular visualizations, generated using PyMOL,¹ result in reduced model performance compared to their 2D counterparts. This trend is consistent across both Isomer and Chirality tasks, suggesting that spatial complexity in graphical representations may hinder model interpretation.

We added this ablation also in the plot Figure R1.

Reviewer Point P 1.6 — 5. I think including safety as its own desired property in the table A.1 would be suitable

Figure R1: Vision Large Language Models (VLLMs) performance for questions related to organic molecules and reactions in MaCBench.

Reply: To clarify that all questions that were labeled “lab protocol” in the previous version are actually related to safety aspects, we changed “lab protocol” to “lab safety” in all instances.

Reviewer Point P 1.7 — 6. I was unable to install the package but all of the questions are there which is the major contribution of the codebase.

Following the instructions to install did not work as some internal functions are no longer working and crashing the program. Cloning the repository and installing it didn’t fix the problem either. I tried to troubleshoot a bit but ultimately decided to leave it. The dataset is all there.

Reply: To address this, we performed a major refactoring of ChemBench (which is the core dependency of MaCBench). The package can be installed with pip (which is tested using continuous integration with GitHub actions), and we provide guidance on how to run the benchmark in online documentation https://lamalab-org.github.io/chembench/getting_started/#how-to-benchmark-on-multi-modal-tasks.

To further simplify reuse independent of the ChemBench codebase, we also provide the dataset on HuggingFace, where leaderboard, results, and data can be found in one collection <https://huggingface.co/collections/jablonkagroup/macbench-collection-67b84c7e346553e4005eb55b>.

REVIEWER 2

Reviewer Point P 2.1 — In their manuscript "Probing the limitations of multimodal language models for chemistry and materials research", Alampara et al. describe a benchmark for evaluation of large language models' (LLMs) capability of analyzing chemical information. The authors subject four well-known LLMs to their benchmark, compare their performance and draw conclusions from their responses. I very much welcome the authors' timely publication, scrutinizing the hype around LLMs and their employment in every possible field of science. I found the article very interesting, and answering a lot of burning questions around the possibilities of LLMs. The benchmark, published for free and open access, has the potential to become a very useful testing tool for LLMs, at least until LLMs become specifically trained on it. I have some questions, comments and suggestions about the reported research and the manuscript itself, but I would strongly recommend the manuscript for publication in this journal.

Reply: *We thank the reviewer for the careful feedback on our manuscript and are glad to read the assessment that our benchmark could become a very useful testing tool—which is the objective of our work.*

Reviewer Point P 2.2 — Can the authors comment about the reproducibility of their benchmark results? As I do not see statistical analysis of any of the responses, I assume each question was asked just once. I find it interesting whether repeating the same question for the same LLM yields different results. If so, deviations between them are as well quite interesting.

Reply: *In the initial version of the manuscript, we only showed the results for one run of the model as we considered the room for variation minimal (since we used greedy decoding). However, we already highlighted some variations due to the refusal triggers (see appendix section "Sensitivity to prompt template").*

In the revised version, we improved the analysis by

- *Making the codebase more robust in handling refusals. This is described in the methods section:*

We implement a comprehensive framework combining regular expression-based detection from large language model (LLM) Guard and a fine-tuned BERT model² to identify potential LLM refusals. This detection pipeline was integrated into our evaluation pipeline, enabling pre-scoring refusal checks. To mitigate refusals, we implemented an interval-based retry mechanism, re-querying the LLM up to n times until a non-refusal response was obtained. For our runs, we retry for a maximum of five times.

- *Rerunning every analysis five times to measure the variance. In all figures in the main text, we now show error bars that illustrate this (minimal) variance across reruns*
- *Including analysis on the sensitivity to the system prompt (see new appendix section "Sensitivity to system prompt")*

Reviewer Point P 2.3 — I find the benchmark baseline somewhat under-described. The authors often mention the "random baseline", but it is not clear to me how it works. For instance, Table A.2 shows the baseline for the extraction of organic reactions schema as 50%. I assume this means randomly extracted schemas are correct 50% of times, but to me such ratio seems very high. I cannot think of

a program which randomly selects correct solvents, temperature and yield from organic reaction schemas half of the time. Maybe if I have the reaction split by components and randomly select one of them, I can reach 50% correct identification for the solvent, but some piece of software would have to do the reaction splitting. Moreover, this holds true only if 50% of components in reactions are solvents. It would be great if the authors could clarify all the "random baseline" selection methods.

Reply: *We now expand on how the random baseline is computed in the appendix*

The random baseline is established by randomly selecting one answer from the available options for multiple-choice questions (MCQs). For example, if there is a MCQs chemistry question asking "Which element has the highest electronegativity?" with options A) Fluorine, B) Oxygen, C) Nitrogen, and D) Chlorine, the baseline would simply pick one letter randomly (e.g., "C"). For numeric questions, we use the mean of all target values within a topic as the prediction. For instance, if there are multiple questions asking for the number of atoms in CIF with answers like 6, 12, and 15, the baseline would calculate the average of all these values (11) and use that as its prediction for every numeric question in that topic.

Reviewer Point P 2.4 — Finally, I find it really surprising that Llama 3.2 90B Vision fails answering all the questions in "Organic Reactions Schema" category, but this might be due to prompt refusals.

Reply: *In the revised version (with the improved code, which also improves refusal handling), Llama-3.2-90B answers half of the questions correctly.*

Reviewer Point P 2.5 — It would be really interesting to see the established pre-AI tools used as a baseline. First, it would make it much easier to understand the concept of the baseline (as indicated above). Second, it would make it much clearer to see how the LLMs rank when compared to the most advanced pre-AI open software tools, such as OPSIN, OSCAR4, OSRA and ChemicalTagger. I understand this suggestion is out of scope for the current manuscript, thus I do not expect it to be answered in the revised manuscript.

Reply: *Following this excellent suggestion, we added such an analysis to the appendix of the revised manuscript. In the new appendix section "Comparison with optical chemical structure recognition tools". Now we include:*

To establish a robust performance evaluation of VLLMs in chemical image analysis, we compared their effectiveness in the hand-drawn molecule recognition task (see Table A.1) against Decimer,^{3,4} a state-of-the-art tool designed explicitly for chemical structure recognition. This comparative analysis serves dual objectives: highlighting the relative strengths of general-purpose VLLMs against domain-specific tools while also assessing whether current VLLM capabilities meet the rigorous performance thresholds required of specialized systems in precision-critical scientific applications.

As shown in Figure R2, the VLLMs (Claude 3.5 Sonnet, Gemini Pro, and GPT-4o) demonstrate superior performance compared to the specialized Decimer model in chemical structure recognition. This suggests that leading VLLMs can surpass technical models for specific cheminformatics tasks like molecule image interpretation. Notably, the error analysis reveals consensus failures between top-performing VLLMs and Decimer. These shared failure cases likely contain structurally complex molecules that present inherent challenges for current recognition systems, as evidenced by consistent performance drops across all models. The correlation in error patterns implies that molecular complexity

Figure R2: Cumulative performance comparison between VLLMs and Decimer on hand-drawn molecular images.

rather than model architecture limitations may be the primary factor in these challenging cases.

Reviewer Point P 2.6 — The authors refer to the tested LLMs as VLLMs (Vision Large Language Models). Why this emphasis of "visual" is needed, when the models described in the paper are also subjected to text-only prompts? It might as well be that I am missing the distinction as I am not an expert in the AI field.

Reply: *We clarify the focus on VLLMs in the revised version of the introduction*

While we have some understanding for text-only LLMs, we still have no understanding for VLLMs that can process images alongside text.

Reviewer Point P 2.7 — When looking at Table A.1, it is not clear to me which of the questions are text-only, image-only and which are text+image. I find this distinction difficult to navigate in the remaining paper as well.

Reply: *We clarify the design of the corpus in the methods section of the revised version of the manuscript*

All questions in our benchmark contain pairs of images and text-based questions. Only some ablation experiments (that are specifically highlighted) contain only text information.

In addition, we added a statement to the caption of the table A.1. (which only contains the main tasks of the benchmark and not the ablations):

All tasks shown in this table consist of an image shown alongside a question in text form.

Reviewer Point P 2.8 — Table A.2 uses bold script to indicate "winners" of each challenge. I find this slightly misrepresenting, as 0.93 gets dominated by 0.97, eclipsing the former and putting it on the same shelf as 0.24. I am not entirely sure how to represent this better, but maybe color gradient could be used, making the best value of maximum color saturation and others at relative? However, authors may disregard this comment altogether, as I do not have a definite solution, and my suggestion might make the table even more complex.

Reply: *We attempted a color coding of the tables but found the result overwhelming, wherefore we stuck to boldening the top entries.*

Reviewer Point P 2.9 — Methodology for calculating the overall performance in Table A.2 needs to be explicit. Is it just the average from all the rows in the table?

Reply: *We added a section in the appendix ("Scoring methodology") that details how the results are computed. The section reads*

For MCQs, a task is considered correct if the Hamming loss is zero, meaning the predicted answer exactly matches the ground truth. For numeric questions, a response is deemed correct if the mean absolute error (MAE) falls within a specified tolerance. The default tolerance is 1%, but for certain question types, such as CIF-Density, CIF-Volume, and some US Patent questions, the tolerance is up to 5% (the tolerance is defined in the curation process). Each correct task receives a score of 1, while incorrect tasks receive a score of 0.

OVERALL & TOPIC SCORES The overall score is calculated as the total number of correct tasks divided by the total number of questions, excluding ablation tasks. Topic-wise scores are computed similarly, with the total number of correct answers in a topic divided by the total number of questions in that topic. For ablation tasks, the same topic-wise scoring method is applied. When tasks or models are combined, their respective scores and standard deviations are averaged.

Reviewer Point P 2.10 — Part of subsection "Scientific terminology" seems missing. The last sentence of this subsection ends abruptly without a period.

Reply: *We completed this sentence in the revised version of the manuscript*

In Appendix A.7, we show that for some questions, large variations in performance can be due to even seemingly minor changes in prompt wording, such as replacing the word "image" with "diagram," "plot," "figure," "photograph," or even omitting it entirely.

Reviewer Point P 2.11 — I have some suggestions how to make the figures easier to read. In Figure 3 part A, there are different ordering of models given in model enumeration and comparison to baseline (i.e., in the upper part cyan color is shown on top, and in the bottom part on bottom). I would suggest retaining the same order everywhere. Moreover, I personally find it difficult to distinguish the colors used for Gemini and Llama models. I understand the visual appeal of the current pastel palette, but please consider using more distant colors. In Figure 3 part B, the same model colors (or very similar to them) are used for drawing group arcs in "Data extraction", "Data interpretation" and so on. This might lead to thinking that these groups are somehow related to particular same-colored LLMs. Please use different colors for unrelated things. Same comments hold true for Figure A.1.

Reply: *We updated the colors in the entire manuscript*

- *We use more distinct colors for Gemini and Llama*
- *The colors we use for models are not used for anything else*
- *The ordering of the legend is now consistent with the ordering of the bars in plots*

Reviewer Point P 2.12 — Please make a stable release of your benchmark and refer to this version in the paper. Since it is hosted on GitHub, I would suggest making a git tag, possibly v0.1.0 or v1.0.0.

Reply: *For the revised version of the manuscript, we made releases of all codes and datasets (see also response to point P 1.7). The datasets are also archived and cited with the corresponding DOI in the data availability section*

To facilitate the benchmarking and reproducibility of our work, we have provided the datasets used in this work on Hugging Face (DOI: 10.57967/hf/4611 and DOI: 10.57967/hf/4612).^{5,6}

The code for running the benchmark is available at <https://github.com/lamalab-org/chembench/> and archived on Zenodo (DOI: 10.5281/zenodo.14935487). Instructions for running the benchmark can be found at https://lamalab-org.github.io/chembench/getting_started/#how-to-benchmark-on-multi-modal-tasks.

Reviewer Point P 2.13 — Figures A.2 and A.3 could use the color legend. I find it tiring needing to jump through the pages to look at the legend in Figure A.1.

Reply: *We added a color legend.*

Reviewer Point P 2.14 — Table A.5 is very interesting and greatly supplements Table A.2, however, I personally find it difficult having to jump from one to another. I would suggest the authors consider merging Table A.5's information in Table A.2, for example by giving the number of rejected questions in brackets next to their correctness value. I am not entirely sure this will not overcrowd the Table A.2, thus I leave the decision to the authors.

Reply: *We attempted to merge the tables but found the merged version overcrowded and hence stuck to the version with two separate tables.*

Reviewer Point P 2.15 — I have some requests for the leaderboard. First, it lacks a clear link to the used benchmark, without it, it is difficult to understand the testing methodology for a passer-by. Next, please use a stable version of the benchmark for the leaderboard. This benchmark version should be clearly indicated in the leaderboard. Finally, please consider versioning the leaderboard itself and making it easier to compare between different versions of the leaderboard. This will become very important with gradual growth of the benchmark (if foreseen), addition, updates and possibly removals of LLMs from the evaluation set.

Reply:

Figure R3: Screenshot of the leaderboard, which we deploy on HuggingFace.

Following this suggestion, we migrated the leaderboard to HuggingFace, where the results are versioned as a HuggingFace dataset and the data are independently versioned as another dataset. The leaderboard is described in more detail in a new section “Leaderboard” in the appendix, which reads:

To summarize the results of MaCBench, we created a leaderboard based on gradio and deployed it on HuggingFace Spaces (Figure R3).⁷ The online leaderboard is available at <https://huggingface.co/spaces/jablonkagroup/MaCBench-Leaderboard>. We version the leaderboard using the HuggingFace dataset, which is version-controlled using git.⁸ Every time there is an update to the leaderboard, the version is bumped, and the user gets to select the version to view in the leaderboard UI.

All resources are compiled in a collection on HuggingFace <https://huggingface.co/collections/jablonkagroup/macbench-collection-67b84c7e346553e4005eb55b>.

Reviewer Point P 2.16 — In the caption for Figure 3, "across all task" should probably be written as "across all tasks" (plural).

Reply: Fixed in the revised version.

Reviewer Point P 2.17 — Abbreviation "AFM" is first introduced in Figure 1, but I feel it should as well be un-abbreviated on its first occurrence in the text, in subsection "Data interpretation".

Reply: We now unabbreviate this instance.

Reviewer Point P 2.18 — In "Data interpretation", "reflexes" should probably be written as "reflections" (seems to be the preferred term in XRD).

Reply: We changed this in the revised version of the manuscript.

Reviewer Point P 2.19 — In the caption for Figure 4, I find the enumeration of VLLMs "(Claude 3.5 Sonnet, ...)" superfluous, as this is evident from the legend.

Reply: We removed the enumeration in the revised version.

Reviewer Point P 2.20 — First sentence of "Synthesis across modalities", "Given that models input visual and textual input ...", I find the multiple usage of word "input" as different parts of speech tricky to read. In the same subsection "four percentage" could probably be better written as "4%, seemingly consistent with the style of the manuscript.

Reply: *Sentence revised to*

Given that models consume visual and textual input in seemingly similar ways, one might expect that the same information is processed in the same way regardless of how it is presented to the model.

We decided to stay with "four percentage point" to make it clear that we mean percentage points and not percent.

Reviewer Point P 2.21 — In Figure 6, "Ablation studys" should probably be written as "Ablation studies".

Reply: Fixed in the revised version.

Reviewer Point P 2.22 — In "A.2 Related work", "Khalighinejad et al. build" should probably be written as "Khalighinejad et al. built" (past tense, consistent within the section).

Reply: Fixed in the revised version.

Reviewer Point P 2.23 — I did not have the possibility to have an in-depth review of the code, but I find it important to note that the code is MIT-licensed. The code needs a stable release, though. I asked the authors to make it in my review text as well.

Reply: *As noted in response to point P 2.12, we made releases of code and data and also archived them.*

REVIEWER 3

Reviewer Point P 3.1 — In the manuscript "Probing the Limitations of Multimodal Language Models for Chemistry and Materials Research," the authors make an effort to explore the application of large language models (LLMs) in the field of chemistry. However, I have several concerns regarding the current state of the manuscript, which I believe need to be addressed before considering it for publication.

Reply: *We thank the reviewer for the careful review of our manuscript.*

Reviewer Point P 3.2 — One of the primary concerns is the reproducibility of the results presented in the manuscript, particularly regarding the use of context windows, which raises questions about the reliability of the findings. The absence of a system prompt to generate informed responses is a significant oversight. Including a system prompt could enhance the model's ability to generate contextually relevant and accurate outputs, thereby improving the reproducibility of the experiments.

Reply: *In the initial version of the manuscript, we did not use a system prompt but included all instructions in the "user" message (similar to prior work such as ChemBench). To analyze the impact of the system prompt, we conducted an ablation, which we describe in the new section "Sensitivity to system prompt" in the appendix.*

Reviewer Point P 3.3 — The manuscript discusses the generation of images but fails to address the general applicability of these images. It is unclear why the authors did not utilize images from published sources, which could have provided a benchmark for comparison. The inclusion of such images would strengthen the validity of the results and offer a clearer understanding of the model's capabilities.

Reply: *Our benchmark does not generate but uses images. We, however, rely on both synthetically generated and naturally occurring images. To clarify this, we now write the following when we describe the corpus in the main text:*

To assess performance in a broad range of settings, we rely on both images we mined from patents but also some we generated from scratch.

Reviewer Point P 3.4 — The definition of the random baseline is ambiguous, particularly concerning the random component. A clear explanation of how the random baseline is established and its role in the benchmarking process is essential for readers to assess the validity of the comparisons made in the study.

Reply: *We added a description of the random baseline definition to the appendix, which reads*

The random baseline is established by randomly selecting one answer from the available options for MCQs. For example, if there's a MCQs chemistry question asking "Which element has the highest electronegativity?" with options A) Fluorine, B) Oxygen, C) Nitrogen, and D) Chlorine, the baseline would simply pick one letter randomly (e.g., "C"). For numeric questions, we use the mean of all target values within a topic as the prediction. For instance, if there are multiple questions asking for the number of atoms in CIF with answers like 6, 12, and 15, the baseline would calculate the average of all these values (11) and use that as its prediction for every numeric question in that topic.

Reviewer Point P 3.5 — The manuscript lacks a discussion on strategies to alleviate the limitations of multimodal LLMs in the context of chemistry tasks. Additionally, the potential benefits of employing modern techniques, such as retrieval-augmented generation, are not explored. These techniques could significantly enhance the model's performance in retrieving accurate data and should be considered in future iterations of the study.

Reply: *We now include a section with recommendations in the appendix (see A.10).*

The degradation in performance for tasks requiring multiple reasoning steps suggests needed improvements in this area:

- **Test time inference:** *Use chain-of-thought approaches to incentive the model to use more test-time compute.⁹*
- **Reasoning Models:** *Leverage recent advances in reasoning models or models with explicit reasoning components, which have shown promise in improving multi-step logical inference. These models specifically designed to strengthen step-by-step reasoning could help address the performance degradation we observed in tasks requiring chained analysis. This could involve fine-tuning on reasoning pathways¹⁰ or reinforcement learning.¹¹*
- **Tool-Augmented Architectures:** *Implement modular architectures where specific components (such as external tools)¹² handle different aspects of multi-step reasoning (e.g., one module for peak identification, another for relative ordering).*

Reviewer Point P 3.6 — Overall, while the manuscript presents an interesting exploration of LLMs in chemistry, the concerns outlined above hinder its suitability for publication in its current form. I recommend that the authors address these issues to improve the clarity, reproducibility, and overall quality of the work. I would not recommend the publication of this manuscript in its current form.

Reply: *To address reproducibility, we performed a major refactoring and moved all data to HuggingFace as outlined in response to point P 2.12.*

REFERENCES

- [1] Schrödinger, LLC, The PyMOL Molecular Graphics System, Version 1.8. 2015.
- [2] ProtectAI.com, Fine-Tuned DistilRoberta-Base for Rejection in the output Detection. 2024; <https://huggingface.co/ProtectAI/distilroberta-base-rejection-v1>.
- [3] Rajan, K.; Zielesny, A.; Steinbeck, C. *Journal of Cheminformatics* **2021**, *13*.
- [4] Rajan, K.; Brinkhaus, H. O.; Zielesny, A.; Steinbeck, C. *Journal of Cheminformatics* **2024**, *16*.
- [5] Lab of Kevin Jablonka at Uni Jena, MaCBench (Revision feb8c43). 2025; <https://huggingface.co/datasets/jablonkagroup/MaCBench>.
- [6] Lab of Kevin Jablonka at Uni Jena, MaCBench-Ablations (Revision c52701f). 2025; <https://huggingface.co/datasets/jablonkagroup/MaCBench-Ablations>.
- [7] Abid, A.; Abdalla, A.; Abid, A.; Khan, D.; Alfozan, A.; Zou, J. *arXiv preprint arXiv:1906.02569* **2019**,
- [8] Lab of Kevin Jablonka at Uni Jena, MaCBench-Results (Revision 0551909). 2025; <https://huggingface.co/datasets/jablonkagroup/MaCBench-Results>.
- [9] Snell, C.; Lee, J.; Xu, K.; Kumar, A. *arXiv preprint arXiv: 2408.03314* **2024**,
- [10] Muennighoff, N.; Yang, Z.; Shi, W.; Li, X. L.; Fei-Fei, L.; Hajishirzi, H.; Zettlemoyer, L.; Liang, P.; Candès, E.; Hashimoto, T. *arXiv preprint arXiv: 2501.19393* **2025**,
- [11] DeepSeek-AI, et al. *arXiv preprint arXiv: 2501.12948* **2025**,
- [12] Ramos, M. C.; Collison, C. J.; White, A. D. *Chemical Science* **2025**,

RESPONSE TO THE REVIEWERS

REVIEWER 1

Reviewer Point P 1.1 — 1. Figure 1 images are grainy, if it's not just my version

Reply: We fixed this in the revised version.

Reviewer Point P 1.2 — 2. Figure 3 fonts are non-standardized and not aligned

Reply: We fixed this in the revised version.

Reviewer Point P 1.3 — 3. On page 6 data extraction is referred to as “the first step of the scientific process”. This isn't right or wrong, but there's some back and forth between the sequential nature of the framework versus the three independent and equally important “pillars”. If data extraction is the “first step”, it might be worth considering explaining why the authors believe that, and how this affects model training. o Considering that these models may be used as a part of larger agent-driven platforms, these semantics will be important to a reader trying to implement their own “AI scientist copilots”. Implying a “starting point” may unintentionally bias a style of design, if it isn't relevant to the study at hand.

Reply: *We changed the phrasing in the text and no longer refer to a sequential nature.*

It now reads:

Interestingly, even for a foundational pillar of the scientific process—data extraction—some models do not perform much better than random guessing.

Reviewer Point P 1.4 — 4. On a similar note, the terms “task”, “task category”, “core scientific tasks”, “the three focus areas”, “ten specialized scientific domains”, “topics”, “subtasks”, etc. need to be standardized. I would suggest explicitly defining each tier and component of the benchmark in the introduction and using those terms consistently throughout the paper. o Note a verb tense misalignment in the caption of figure 3A (“across all task”)

Reply: *We now use the following terms:*

- *Task: A single prompt template containing multiple questions.*
- *Topic: A collection of tasks related to the same topic.*
- *Focus Area: Encompassing multiple topics, these are data extraction, data interpretation, and in-silico & lab experiments.*

In the revised text, we define these three key terms in the “The MaCBench framework” subsection.

*To ensure clarity and consistency throughout this paper, we will use the following terms: A **task** refers to a single prompt template containing multiple questions. A task can either be a multiple-choice question or a numeric answer question. The current corpus has 779 MCQ questions and 374 numeric answer questions. A **topic** is a collection of tasks related to the same topic (one topic can have different types of tasks related to that topic; for example, XRD can have multiple tasks related to identifying peak positions, and then another set of tasks related to ordering peak positions in ascending/descending order). The three overarching **focus areas** are data extraction, data interpretation, and experiments, each encompassing multiple topics.*

We replaced all other instances the reviewer mentioned with one of these three terms.

Reviewer Point P 1.5 — o The error bars of figure 3A are barely visible

Reply: We now draw error bars in black to make them better visible.

Reviewer Point P 1.6 — 5. It may be worthwhile mentioning that some of the limitations are specific to VLLMs and not AI-driven experimentation (specifically as a preface in the Core Reasoning Limitations section). That is, it would be unlikely to directly have a VLLM interpret some experimental spectra from the image if the digital information was available and could be preprocessed. o In the context of having a VLLM “read” (e.g., an android reading off a computer), such an evaluation makes sense. o But in a cloud-based system where data is uploaded directly from an analytical instrument and digitally processed before digested by a central AI (which may be driven by a (V)LLM agent), its performance in identifying peaks directly from an image is less relevant. o In general, it would be good to temper both positive and negative claims with the (in-lab) application context envisioned by the authors to better calibrate the reader’s interpretations and take-aways

Reply: *We added a paragraph to add context*

Looking forward, it is also important to note that for future workflows, with advanced data management¹ or self-driving labs² some of the tested multimodal integration abilities will be less important as data will directly be available in a machine-actionable form instead of requiring parsing from an image.

Reviewer Point P 1.7 — 6. With respect to multistep reasoning, I think the audience would now be curious about the newer deepseek r1 and GRPO strategies, if the authors are able to incorporate. o Admittedly, it’s possible by the time the authors could, there might already be a new advancement in the field.

Reply: *At this point, there is no publicly available reasoning-VLLM and training a new one is outside the scope of our article. However, our leaderboard enables new models to be added easily once they are published, and we plan to do so.*

Reviewer Point P 1.8 — 7. Figure 5 might say % correct in the x axis?

Reply: *To clarify the figure, we performed the following changes in the revised version:*

- *Enlarged dots and thinner lines to emphasize the binary character of the measurements.*
- *Clarified caption that now reads:*

VLLM performance as a function of number of search hits. The plots compare four leading VLLMs across different crystallographic tasks: a. atomic species identification, b. crystal system classification, c. density calculation, and d. crystal symmetry determination. For each property, log-scale Google hit counts are plotted against the binary correctness (correct/incorrect) of model responses, with lines serving as visual aids only, revealing correlations between answer accuracy and the prevalence of information in online sources. Higher hit counts for correct answers suggest models may not solely rely on reasoning in their responses to crystal structure analysis tasks.

Reviewer Point P 1.9 — 8. Was there any assessment performed on the model performance with respect to the original image quality (not the scaled resolution,

Figure R1: Comparison of model robustness to image noise. Subplot shows the fraction of correctly answered questions as a function of increasing noise levels (0–0.99) for three tasks: Electronic Structure, Handdrawn Molecules, US Patent Plots, and XRD Peak Position. Images were degraded by adding Gaussian noise with zero mean and varying standard deviation (`noise_level`). The noise was generated using `np.random.normal(0, noise_level*255, image_shape)` was added directly to pixel values, then clipping to the valid 0–255 range. Each data point represents the average of three independent trials (with noise created using different seeds), and the error bars indicate the standard deviation of these averages.

but rather a study seeing how different amounts of noise in the source image can affect results?)

Reply:

We now added an ablation study to assess model performance with noise in images by adding Gaussian noise to pictures on a few topics. We added the results to a new subsection in the Appendix:

Sensitivity to noise in image

The noise ablation analysis reveals significant differences in model robustness across tasks and models. While some models maintain relatively stable performance regardless of noise (Llama 3.2-90B on Electronic Structure), others exhibit threshold effects where performance collapses after certain noise levels (Claude 3.5 Sonnet on Electronic Structure). GPT-4o shows the steepest performance drop in all the tasks, while Llama 3.2-90B degrades more gradually. The almost stable performance of XRD might be due to distinctive peak patterns at specific positions, which make them recognizable even when additional noise is introduced.

Reviewer Point P 1.10 — 9. In terms of the desired properties enumerated in A1, there is a point about experimental planning. While it may be too broad of a scope to include as experiments in the current work, a discussion on model ability to directly interface with robotics may be appropriate. o This was just published <https://pubs.acs.org/doi/10.1021/jacs.4c17738> o I believe ORGANA has been published o <https://www.nature.com/articles/s41467-024-54457-x>

Reply:

We added those references and a short discussion

This is also relevant for integration with robotic setups.^{3–5} While this will require agentic abilities (not covered in our benchmark), it might also require the robotic setup to reason about images taken of the current state of the experiment

Reviewer #1 (Remarks on code availability):

Working as described, but I would suggest adding the instructions of adding an API key to your environment within the repository readme .

Reply: *We added this to the README now.*

REVIEWER 2

Reviewer Point P 2.1 — I would like to thank the authors for considering all my questions, suggestions and comments. The authors have greatly improved the manuscript, making it much easier to grasp the most important details of their study. However, I feel that couple of my original points have been answered only partially, thus I would like to request the authors for additional revision.

Reviewer Point P 2.2 — First, in the initial manuscript I completely overlooked the fact that the LLMs were solving the multiple-choice questions (MCQs). Only after the revision of the initial manuscript it came to my view that this was the case, in contrast to answering open-ended questions. It is still not clear to me whether all the questions were MCQs, or just part of them. I think this attribute of the study needs to be stressed even more, and has to be introduced as early as possible. Otherwise a reader might get an overly optimistic view of LLMs (like I did).

Reply: *We now stress this when introducing the corpus
Early in the text, we now introduce a sentence that states,*

*A task can either be a multiple-choice question or a numeric answer question.
The current corpus has 779 MCQ questions and 374 numeric answer questions.*

Reviewer Point P 2.3 — Correctness of LLMs' choices in MCQs vastly depends on the number of options, how close they are one to another and how well they sample the response space. Take a simple optical character recognition test for example: a recognizer is much more likely to choose "O" as correct answer if given "O", "I" and "X" to choose from than compared to "O", "o", "Q", "C", "D", or the whole alphabet. What is more, there is a need to know the correct answer as one of the possibilities apriori. Thus I expect a detailed discussion on how the options for MCQs were selected.

Reply: *We have now added a table in the Appendix describing how options have been selected for each task.*

Table R1: Details on how options were created for MCQ questions

Topic	Option Selection Description
Data Extraction	
Hand-drawn Molecules	Options consist of 4 choices as SMILES: the correct molecule and 3 distractors randomly sampled from the hand-drawn molecule dataset.
Organic Chemistry	
Chirality	Questions to identify chiral centers options are 0, 1, 2, 3, or more. Options in questions about determining the spatial orientation relative to the molecular plane are about describing positions (toward the viewer, away from the viewer) and relationships to other atoms (closer/further due to electrostatic attraction/steric repulsion).
Isomers	Options consistently include all possible isomeric relationships: chain isomers, functional group isomers, regioisomers, enantiomers, diastereomers, conformers, identical molecules, or no isomeric relation.
Organic Molecules	Hand-crafted options designed to test understanding of molecule rendering and IUPAC nomenclature rules with plausible distractors that require precise knowledge to differentiate.
Organic Reaction Schema	Four carefully designed options per question that require detailed analysis of reaction schemas, with distractors that challenge elimination.
Organic Reaction Schema without SMILES	Four options per question requiring visual analysis of reaction schemes without SMILES notation, with challenging distractors.
Tables and Plots	
Composition Tables	Questions include binary (yes/no) questions and multiple-choice questions with carefully designed distractors targeting common misinterpretations of tabular data.
US Patent Figures	Manually-crafted options requiring detailed analysis of patent diagrams with plausible alternatives targeting common analytical errors.
US Patent Plots	Manually-crafted options requiring detailed interpretation of data visualizations from patents, with distractors targeting common analytical errors.
In silico and lab experiments	
Lab QA	
Lab Safety	Options crafted to test fundamental laboratory safety knowledge with plausible alternatives that require genuine understanding.
Lab Safety Comparison	Options designed to test comparative analysis of laboratory safety procedures with distractors that challenge easy eliminations of methodological differences.
Lab Equipments	Four options per question drawn from a set of 10 possible laboratory equipment items, requiring precise identification based on form and function.
CIF QA	
Crystal Structure Symmetry	Four options consisting of the correct space group and three plausible alternatives randomly selected from the 230 possible crystallographic space groups.
Crystal System	Options always include all seven crystal systems (cubic, tetragonal, orthorhombic, hexagonal, trigonal, monoclinic, and triclinic).

Data Interpretation	
AFM Image Analysis	Manually-crafted options requiring detailed interpretation of atomic force microscopy images, with distractors that challenge easy eliminations.
Adsorption Isotherm Adsorption Isotherm Capacity Comparison	Options list the three MOF frameworks being compared, with the correct answer identifying the framework with either the highest or lowest adsorption capacity at specified conditions.
Adsorption Isotherm Capacity Order	Options present different ordering sequences of four MOFs, with only one correctly representing the ascending or descending sequence of adsorption capacities.
Adsorption Isotherm Capacity Value	Options include the correct numerical uptake value plus three plausible alternatives randomly sampled within ± 1 mol/kg of the correct answer, all formatted to two decimal places.
Adsorption Isotherm Henry Constant Comparison	Options list three MOF frameworks selected for distinguishable Henry constants, with the correct answer identifying the one with the highest or lowest value.
Adsorption Isotherm Henry Constant Order	Options present different ordering sequences of four MOFs, with only one correctly showing the ascending or descending order of Henry constants.
Adsorption Isotherm Strength Comparison	Options list four MOF frameworks, with the correct answer identifying the one with either the highest or the lowest adsorption strength at low pressure (0.01 bar).
Adsorption Isotherm Strength Order	Options show different possible ordering sequences of MOFs, with only one correctly representing the ascending or descending order of adsorption strengths at low pressure.
Adsorption Isotherm Working Capacity Comparison	Options list three MOF frameworks, with the correct answer identifying the one with the highest or the lowest working capacity between two specified pressures.
Adsorption Isotherm Working Capacity Order	Options present different ordering sequences of three MOFs, with only one correctly showing the ascending or descending order of working capacities between specified pressures.
Adsorption Isotherm Working Capacity Value	Options include the correct numerical working capacity plus three plausible alternatives randomly sampled within a range from zero to 1 mol/kg above the correct value, formatted to two decimal places.
Electronic Structure	Options consistently include three fundamental band structure classifications: direct bandgap, indirect bandgap, and metallic (no bandgap).
NMR and MS Spectra	Four expert-crafted options per question require detailed interpretation of spectroscopic data, with distractors targeting common analytical errors in peak assignment or structure determination.
XRD QA XRD Pattern Matching	Options present different combinations of crystal structure types (simple cubic, body-centered cubic, face-centered cubic) for two XRD patterns displayed side-by-side, requiring identification of the correct structure for each pattern.
XRD Pattern Shape	Four options categorizing two side-by-side XRD patterns as either crystalline or amorphous (Both crystalline, Both amorphous, Pattern 1 crystalline/Pattern 2 amorphous, or Pattern 1 amorphous/Pattern 2 crystalline).
XRD Peak Position	Four options including the correct 2θ value of the most intense peak plus three plausible alternatives randomly sampled from the pattern's 2θ range, requiring precise peak position identification.
XRD Relative Intensity	Four options presenting different possible orderings of peak positions (2θ values) arranged by intensity, with notation indicating ascending (<) or descending (>) order relationships.

Reviewer Point P 2.4 — I am grateful for the authors to responding to my suggestions regarding versioning. Now ChemBench and all the data on the leaderboard is versioned. What I miss is the version of MacBench in its Hugging Face page. For MacBench-Results I see "1.0.0" under "Split" tab under "Dataset Viewer", but the same does not hold for MacBench. GitHub page for MacBench does not have tags as well.

Reply: *While both leaderboard and question datasets utilize Hugging Face Datasets for versioning, their procedures differ slightly to ensure long-term maintainability. For the leaderboard, where the schema remains constant, we employ separate splits within the same dataset. This approach enables dynamic selection and viewing of leaderboard tables within the MaCBench Leaderboard space, as per reviewer suggestions. Conversely, the question dataset is susceptible to schema changes (e.g., new topic additions or subset modifications). Thus, we employ versioning through distinct datasets within the MaCBench collection. To maintain traceability, the Leaderboard UI hyperlinks each dataset version to its corresponding question dataset, as illustrated in Figure R2.*

We now also clarify the versioning in the Leaderboard section in the appendix, which states,

Note that Leaderboard versions are managed through distinct splits within a single HuggingFace dataset, leveraging its stable schema. However, question dataset versions, due to potential schema changes, are maintained as separate Hugging Face datasets within the MaCBench collection.

Figure R2: Screenshot illustrating dataset versioning.

We also made a release for the MacBench repository on GitHub <https://github.com/lamalab-org/macbench/releases/tag/v1.0.0>.

Reviewer Point P 2.5 — What is more, the manuscript should include the versions of all the software (I assume ChemBench) and all the data (MacBench) used to arrive to its conclusions. Please also add these versions to the text. I am aware that DOIs are provided, however, just by looking at DOIs (and even their landing pages) it will be difficult for the future reader to compare the versions (for example, to check whether major versions are the same etc).

Reply:

We have now added this detail in the revised version of the manuscript

To facilitate the benchmarking and reproducibility of our work, we have provided the datasets (version 1.0.0 for the MaCBench dataset) used in this work on HuggingFace (DOI: 10.57967/hf/4611 and DOI: 10.57967/hf/4612).^{6,7}

The code for running the benchmark is available at <https://github.com/lamalab-org/chembench/> and archived on Zenodo (DOI: 10.5281/zenodo.14935487). We used version v0.3.0 for this study. Instructions for running the benchmark can be found at https://lamalab-org.github.io/chembench/getting_started/#how-to-benchmark-on-multi-modal-tasks.

REFERENCES

- [1] Jablonka, K. M.; Patiny, L.; Smit, B. *Nature Chemistry* **2022**, *14*, 365–376.
- [2] Tom, G. et al. *Chemical Reviews* **2024**, *124*, 9633–9732.
- [3] Darvish, K.; Skreta, M.; Zhao, Y.; Yoshikawa, N.; Som, S.; Bogdanovic, M.; Cao, Y.; Hao, H.; Xu, H.; Aspuru-Guzik, A.; Garg, A.; Shkurti, F. *Matter* **2025**, *8*, 101897.
- [4] Ruan, Y.; Lu, C.; Xu, N.; He, Y.; Chen, Y.; Zhang, J.; Xuan, J.; Pan, J.; Fang, Q.; Gao, H.; Shen, X.; Ye, N.; Zhang, Q.; Mo, Y. *Nature Communications* **2024**, *15*.
- [5] Song, T. et al. *Journal of the American Chemical Society* **2025**,
- [6] Lab of Kevin Jablonka at Uni Jena, MaCBench (Revision feb8c43). 2025; <https://huggingface.co/datasets/jablonkagroup/MaCBench>.
- [7] Lab of Kevin Jablonka at Uni Jena, MaCBench-Ablations (Revision c52701f). 2025; <https://huggingface.co/datasets/jablonkagroup/MaCBench-Ablations>.